# The Cross-Talk between Thrombosis and Inflammatory Storm in Acute and Long-COVID-19: Therapeutic Targets and Clinical Cases

**DOI:** 10.3390/v13101904

**Published:** 2021-09-23

**Authors:** Domenico Acanfora, Chiara Acanfora, Marco Matteo Ciccone, Pietro Scicchitano, Alessandro Santo Bortone, Massimo Uguccioni, Gerardo Casucci

**Affiliations:** 1Department of Internal Medicine, San Francesco Hospital, Viale Europa 21, 82037 Telese Terme, Italy; acanforachiara@gmail.com; 2Department of Biotechnological and Applied Clinical Sciences, University of L’Aquila, 67100 L’Aquila, Italy; 3Section of Cardiovascular Diseases, Department of Emergency and Organ Transplantation, School of Medicine, University of Bari, 70124 Bari, Italy; marcomatteociccone@gmail.com (M.M.C.); piero.sc@hotmail.it (P.S.); 4Division of Cardiac Surgery, Department of Emergency and Organ Transplantation, University of Bari, 70124 Bari, Italy; alessandro.bortone@gmail.com; 5Cardiology Unit, San Camillo Hospital, 00152 Rome, Italy; muguccioni@hotmail.com

**Keywords:** DOACs, COVID-19, anticoagulants, LMWH, PARs, Long-COVID-19

## Abstract

Severe acute respiratory syndrome coronavirus-2 (SARS-CoV-2) commonly complicates with coagulopathy. A syndrome called Long-COVID-19 is emerging recently in COVID-19 survivors, characterized, in addition to the persistence of symptoms typical of the acute phase, by alterations in inflammatory and coagulation parameters due to endothelial damage. The related disseminated intravascular coagulation (DIC) can be associated with high death rates in COVID-19 patients. It is possible to find a prothrombotic state also in Long-COVID-19. Early administration of anticoagulants in COVID-19 was suggested in order to improve patient outcomes, although exact criteria for their application were not well-established. Low-molecular-weight heparin (LMWH) was commonly adopted for counteracting DIC and venous thromboembolism (VTE), due to its pharmacodynamics and anti-inflammatory properties. However, the efficacy of anticoagulant therapy for COVID-19-associated DIC is still a matter of debate. Thrombin and Factor Xa (FXa) are well-known components of the coagulation cascade. The FXa is known to strongly promote inflammation as the consequence of increased cytokine expression. Endothelial cells and mononuclear leucocytes release cytokines, growth factors, and adhesion molecules due to thrombin activation. On the other hand, cytokines can activate coagulation. The cross-talk between coagulation and inflammation is mediated via protease-activated receptors (PARs). These receptors might become potential targets to be considered for counteracting the clinical expressions of COVID-19. SARS-CoV-2 is effectively able to activate local and circulating coagulation factors, thus inducing the generation of disseminated coagula. LMWH may be considered as the new frontier in the treatment of COVID-19 and Long-COVID-19. Indeed, direct oral anticoagulants (DOACs) may be an alternative option for both early and later treatment of COVID-19 patients due to their ability to inhibit PARs. The aim of this report was to evaluate the role of anticoagulants—and DOACs in particular in COVID-19 and Long-COVID-19 patients. We report the case of a COVID-19 patient who, after administration of enoxaparin developed DIC secondary to virosis and positivity for platelet factor 4 (PF4) and a case of Long-COVID with high residual cardiovascular risk and persistence of blood chemistry of inflammation and procoagulative state.

## 1. Introduction

The coagulation pathway is a cascade of events that leads to hemostasis in order to heal wounds and prevent spontaneous bleedings. Intrinsic and extrinsic pathways converge to generate and activate fibrin. Severe acute respiratory syndrome (SARS-CoV)-2—responsible for coronavirus disease 2019 (COVID-19) pandemic—seems to evolve with severe coagulopathy [1]. Disseminated intravascular coagulation (DIC) is a common complication in sepsis [2] and can be associated with high death rates in COVID-19 patients [3,4,5,6]. Therefore, DIC and subsequent alterations in coagulation processes in severe COVID-19 may be a hallmark for clinical degeneration in COVID-19 [5]^.^ Sepsis is effectively able to promote the coagulation process: the first step is the generation of the tissue factor (TF), which leads to the activation of the extrinsic pathway of the coagulation process; meanwhile, the increase in plasminogen activator inhibitor-1 (PAI-1) inhibits the fibrinolysis cascade. Tang et al. also reported the down-regulation of Protein C in sepsis: this resulted in the reduction of the anticoagulation properties of Protein C pathway in sepsis [5]. Recent advances in comprehending the pathogenic mechanisms of coagulation and fibrinolysis in sepsis may have therapeutic implications. Recombinant human activated protein C (rhAPC; drotrecogin-alfa activated) was the only pharmacologic therapy that demonstrated reduction in mortality in adults with severe sepsis [7], but no data are available about its administration in patients with COVID-19. Food and Drug Administration has revoked the use of drotrecogin-alfa (activated) due to a significant increase in bleeding and poor efficacy [8]. Systemic corticosteroid administration and long-term bed rest also increase the risk of venous thromboembolism (VTE) in patients with COVID-19. Theoretically, administration of anticoagulants (such as heparin) can be considered as a good therapeutic option in patients with COVID-19, though their efficacy and safety are still a matter of debate in such a context [9]. Thrombin and Factor Xa (FXa) are well-known components of the coagulation cascade. The FXa is known to strongly promote inflammation as a consequence of the increase in cytokine expression. Endothelial cells and mononuclear leucocytes release cytokines, growth factors, and adhesion molecules due to thrombin activation. On the other hand, cytokines can activate coagulation. The cross-talk between coagulation and inflammation is mediated via protease-activated receptors (PARs). These receptors might become potential targets to be considered for counteracting the clinical expressions of COVID-19. COVID-19 is effectively able to activate local and circulating coagulation factors, thus inducing the generation of disseminated coagula [10]. Recently, the International Society of Thrombosis and Hemostasis (ISTH) identified an earlier phase of sepsis-associated DIC, named “sepsis-induced coagulopathy” (SIC), which can benefit from anticoagulants. [11,12] The patients who meet the diagnostic criteria of SIC may benefit from anticoagulant therapy [12]. Given the common use of anticoagulants worldwide, international guidelines about the management of these drugs in patients with COVID-19 are needed. The aim of this report was to evaluate the possible clinical effects of anticoagulants in COVID-19 patients and the role of direct oral anticoagulants (DOACs) as alternative therapeutic options to heparin in relation to their inhibitory effects on PARs. In particular, a dedicated focus was set on the withdrawal of DOACs and substitution with heparins in high-risk patients (i.e., elderly, atrial fibrillation, coronary artery disease, peripheral arterial disease, heart failure, VTE, cancer, etc.) with known or suspected COVID-19.

## 2. COVID-19 and Older Adults with Comorbidities

Older people are at higher risk for severe and fatal forms of COVID-19 due to their frailty and comorbidities [3,13,14]. Experience from Italy shows a median age at death of 79 years for men and 82 for women [15]. The COVID-19 disease has been defined as a “pandemic” since 11 March 2020 by the World Health Organization (WHO). As of April 2nd, the death rate was double that of Severe Acute Respiratory Syndrome (SARS, 2002–2003) and Middle-East Respiratory Syndrome (MERS, 2013) [16]. This pandemic seems to expand at an exponential rate, doubling the positive cases every 43 h. New COVID-19 populations are generally liable, but elderly people with underlying diseases are more susceptible. Diabetes, hypertension, obesity, cardiovascular disease, and cerebrovascular disease are the most important comorbidities implied in degeneration of clinical conditions of patients with COVID-19 [13,14]. Elderly individuals showed the most severe SARS-CoV-2 phenotypes, were more frequently admitted to the intensive care unit (ICU) and demonstrated higher mortality rates [15,16,17,18]. Yang et al. found that 52% of their COVID-19 population was older than 60 years old with higher prevalence in chronic medical illnesses [19]. Indeed, “Inflammaging”, i.e., a sort of dysregulated immune response with exacerbated inflammatory and depressed immunologic components, is a typical feature of aging and might make the elderly more vulnerable to COVID-19, mainly by promoting the cytokine storm [20]. Interestingly, the resistance of the bat to the toxic effect of COVID-19 is explained by a well-balanced immune response with mild inflammatory component [21]. Furthermore, the proportion of fat mass increases with age [22]. Thus, geriatric patients might be at greater risk of cytokine storm also if their BMI does not fall within the obesity range. Indeed, adipose tissue has a proinflammatory effect [23]. Appropriate medical management of cardiovascular comorbidities, including the correct use of anticoagulants, was supported to possibly promote favorable effects on the prognosis of patients with COVID-19 pneumonia [24].

## 3. Uncertain Effects of Anticoagulants in COVID-19

The virosis induced by SARS-CoV-2 promotes endothelial dysfunction, through direct invasion of endothelial cells [25], which in turn can induce excessive thrombin generation and fibrinolysis shutdown, which are conditions able to promote the activation of coagulation [10,25]. Endothelial dysfunction leads to microvascular dysfunction with consequent vasoconstriction, ischemia, inflammation, edema and a procoagulative state [25]. The reduction in Oxygen (O2) plasma levels in severe COVID-19 may induce a prothrombotic status due to the activation of a hypoxia-inducible transcription factor-dependent signaling pathway [10]. All these conditions may lead to occlusion and micro-thrombosis in pulmonary small vessels of severe COVID-19 patients, as outlined in autoptic reports [4]. Therefore, early administration of anticoagulants in COVID-19 was suggested in order to improve the outcome of patients, although exact criteria for their application were not well-established [26]. Low-molecular-weight heparin (LMWH) was commonly adopted for counteracting DIC and venous thromboembolism (VTE), due to its pharmacodynamics and anti-inflammatory properties [27]. The prophylactic dose of LMWH was used by Tang et al. [28] and such administration was related to low and mild bleeding complication rates. Indeed, DIC may develop due to the occurrence of reduction in platelet count and prolongation of pro-thrombin time (PT). These alterations are related to an increased mortality rate. For this reason, anticoagulation may be challenging. ISTH proposed new SIC criteria to manage anticoagulation in such a context, based upon a dedicated score which had been previously validated [12]. Nevertheless, severe COVID-19 patients poorly met SIC criteria, and therefore should not theoretically undergo anticoagulant treatment. For example, platelet count cannot be considered as a sensitive marker for coagulopathy in severe COVID-19 pneumonia; therefore, one of the SIC criteria—platelets depletion—could not be considered for the final decision making [28]. Conversely, markedly elevated D-dimers—indirect markers for coagulation activation—suggest possible benefits from heparin therapy. Nevertheless, the activation of coagulation may reduce pathogen dissemination and invasion [29]. Therefore, anticoagulation may be dangerous in patients without significant coagulopathy, thus explaining the occurrence of higher mortality rate in heparin users as compared to controls with lower D-dimer plasma levels [28]. LMWH are nowadays considered as the new frontier in counteracting the advancing of COVID-19. Other therapies have not been considered yet. Physicians should consider anticoagulation with heparin or LMWH as not beneficial to unselected patients; only those meeting SIC criteria or with markedly elevated D-dimer may be considered for possible anticoagulation with LMWH [28,30].

## 4. Potential Benefits of DOACs in COVID-19

The blood coagulation cascade is initiated by tissue factor (TF)/Factor VIIa (TF/FVIIa) complex and conveys to the generation of Xa factor (FXa) first, and then thrombin (II Factor) [31,32,33]. Thrombin is a proteolytic enzyme of coagulation cascade able to cleave fibrinogen into fibrin and activate platelets. Aberrant activation of blood coagulation system can contribute to the degeneration of different pathologies, such as COVID-19 pneumonia (Figure 1).

Vessel thrombosis leads to ischemia and, if the occlusion is prolonged, to the necrosis of tissues and organs, while degradation products may enhance inflammation into vessels mainly due to the activation of specific proinflammatory receptors such as protease-activated receptors (PARs) [34]. Four PARs (PAR1–4) are known, which are ubiquitously expressed [31,34]. PARs typically serve as both receptor and ligand: proteolytic cleavage by an activated coagulation factor leads to exposure of a neoamino terminus which activates the receptor itself, thus leading to widespread intracellular signaling [34]. PAR’s 1, 3 and 4 are usually activated by thrombin, while PAR-2 by the TF/FVIIa complex, factor Xa, and trypsin [31]. Furthermore, PAR-1 acts as the receptor for TF/FVIIa complex and factor Xa. Thrombin activates PAR1, PAR3, and PAR4, whereas FXa primarily activates PAR2 and PAR1 [31,34]. Direct oral anticoagulants (DOACs) specifically inhibit FXa or thrombin: apixaban, edoxaban, and rivaroxaban are inhibitors for FXa, while dabigatran etexilate is a thrombin inhibitor [31]. The role of this compound in sepsis has not been evaluated yet. Posma et al. evaluated the role of each proteases of the coagulation cascade, particularly FXa and thrombin, and PARs in different mouse models of inflammatory diseases, including virosis (H1N1 strain of influenza A virus) [35]. The blood coagulation cascade may be activated during virosis, thus provoking DIC [31]. Inhibition of the TF/FVIIa complex reduced the inflammatory background of the virosis and the mortality rate in primate model infected by Ebola virus [36]. Indeed, the roles of PAR1 and PAR2 in mouse models of viral infections are controversial [37]. PAR1 inhibition protected mice against respiratory syncytial virus and human metapneumovirus infection [38]. In vitro studies with human A549 cells showed that PAR1 inhibition reduced the replication of respiratory syncytial virus and human metapneumovirus infection [35]. Animal studies showed a significant reduction in inflammation in H1N1 infection when rivaroxaban was administered, dabigatran etexilate being unable to reproduce such results [35]. Ellinghaus et al. demonstrated a reduction in thrombin-induced expression of pro-inflammatory genes in human endothelial cells after administration of rivaroxaban and dabigatran. Indeed, dabigatran showed a biphasic reaction compared to rivaroxaban as it transiently increased the expression of pro-inflammatory genes at concentrations below the minimal inhibitory effective concentrations [39]. Nevertheless, poor data are available for translating these results in clinical practice, while no studies have evaluated the effects of DOACs in COVID-19.

## 5. Clinical Case

A 45-year-old male has been hospitalized for SARS-CoV-2 nasopharyngeal RNA swab positivity for 10 days due to fever, breathlessness, and peripheral oxygen desaturation. He was reported no to have any noteworthy diseases, except for fracture surgery of the right tibia in January 2020. Vital signs on admission were as follows: blood pressure, 150/90 mm Hg; heart rate, 96 beats/min; respiratory rate, 28/min; body temperature, 39.8 °C; oxygen saturation through Venturi mask at 12 L/min 57%; pH 7.49; arterial carbon dioxide partial pressure 36 mmHg; arterial oxygen partial pressure 57 mmHg; PaO_2_/FiO_2_ 114; Modified Early Warning Score (MEWS) 4 [40]; the severity CT score index was 4 [41] (Figure 2).

The patient was treated with ceftriaxone (2 gr ev od), clarithromycin (500 mg ev bid), methylprednisolone (40 mg ev bid), enoxaparin (4000 UI 0,4 subcutaneous bid), furosemide (20 mg ev od), paracetamol (1 g ev every 4 h). Worsening of patient’s condition was observed on the second day of hospitalization (MEWS 8). High flow non-invasive ventilation was started to FiO2 60%. Laboratory data showed that the disseminated intravascular coagulation (DIC Score = 6) was suggestive of overt DIC [42] (Table 1). Anti-platelet factor 4 (PF4) was present and enoxaparin was stopped. Arterial and venous Doppler ultrasound of the lower limbs and supraortic vessels showed no abnormalities. Abdominal ultrasound, CT scan of the brain and MR angiography of intra- and extra-cranial vessels did not show relevant signs. Based on the European Society of Cardiology Guidance (CV disease/COVID-19 2020) [26], Direct Oral Anticoagulant was started (Rivaroxaban 15 mg bid). On the seventh day, coagulation parameters were normalized (DIC score = 0) and clinical conditions improved with MEWS score 0. The patient was discharged and went home being apyretic and negative SARS-CoV-2 nasopharyngeal RNA swab, with home therapy: Rivaroxaban 15 bid for two weeks followed by Rivaroxaban 20 mg od and Prednisone 25 mg od for seven days. DIC is an acquired syndrome whose main characteristic is an alteration in blood coagulation. Insults or injuries with a high risk of leading to DIC can be both infectious and non-infectious. In the patient observed in our COVID-19 area, we obtained a rapid remission of symptoms and normalization of hematochemical parameters, in particular those relating to DIC, after withdrawal of enoxaparin. The patient experienced exposure to enoxaparin at tibial fracture surgery, and anti-PF4 was also present. The pathophysiology of DIC in our patient is likely to be related to both SARS-CoV-2 infection (Figure 1) and the presence of antiplatelet antibodies (Figure 3).

The choice to treat the patient with rivaroxaban derives primarily from the absolute contraindication to the use of heparins; AVKs have a slow onset of anticoagulant effects and require continuous monitoring of INR; for dabigatran, apixaban, and edoxaban, evidence of efficacy in ill patients is lacking; for rivaroxaban there is evidence of a reduction in fatal and major thromboembolic events in medically ill patients [43]. A timely diagnosis is therefore the best way to reassure patients. Blood levels of fibrinogen, d-dimer, and platelet counts can be useful as more than reliable biological markers of an ongoing DIC or a probable onset of it.

## 6. Long-COVID-19

Patients discharged from hospital after acute COVID-19 had an increased risk of multiorgan dysfunction, readmission, and mortality [44]. Recent joint guidelines proposed by the National Institute for Health and Care Excellence (NICE), the Scottish Intercollegiate Guidelines Network (SIGN), and the Royal College of General Practitioners (RCGP) have divided COVID-19 infection into 3 phases—‘Acute COVID-19’ (signs and symptoms of COVID-19 infection up to 4 weeks), ‘ongoing symptomatic COVID-19’ (from 4 weeks up to 12 weeks), and ‘post-COVID-19 syndrome’ (when signs and symptoms continue beyond 12 weeks). The term ‘Long-COVID-19’ is given to the signs and symptoms that continue or develop after the ‘acute COVID-19’ phase and include both ‘ongoing symptomatic COVID-19’ and ‘post COVID-19 syndrome’ [45]. In addition to the persistence of symptoms, it is also possible to detect abnormalities of chest radiographs and biomarkers [46]. Interestingly, in Long-COVID-19 patients, it is possible to detect a prolonged elevation of D-dimer, regardless of the inflammatory indices and the severity of the acute phase. In these patients with persistent elevation of the D-dimer, there is an increase in serious thromboembolic complications [47].

## 7. Materials and Methods and Preliminary Results

To characterize the clinical picture, laboratory findings and prognosis of patients with long-COVID-19 were reported preliminary data of patients belonging to our Institute. We consecutively enrolled 50 patients admitted to our Institute from 1 May 2021 to 30 June 2021 for symptoms characterized by dyspnea, fatigue, cough, headache, loss of appetite, myalgia. Of the patients enrolled, thirty recovered from COVID-19. Table 2 summarizes the clinical characteristics of the study population.

Length of acute COVID-19 was 23.1 ± 8 (range 11–49) days. During the acute phase of COVID-19, according to Severity of COVID-19 WHO Clinical Classification [48], 21 patients were classified as Mild/Moderate and 9 as Severe/Critical. On the Post-COVID-19 Functional Status Scale [49] 7 (23.3%) patients reported No/Negligible functional limitations for, 6 (20%) Slight Functional Limitations and 17 (56.7%) Moderate/Severe Functional Limitations. No differences were found in demographics, medical history and vital signs in patients with Long-COVID-19 compared to no COVID-19 patients. Echocardiographic findings showed that left ventricular ejection fraction was lower in Long-COVID-19 patients. Table 3 summarizes laboratory data of Long-COVID-19 patients and no COVID-19 patients; inflammatory parameters and coagulation pathway were higher in Long-COVID-19 patients.

## 8. Clinical Case

71-year-old man, diabetic, hypertensive, hypercholesterolemic, chronic ischemic heart disease, previous acute coronary syndrome (ACS) with ST elevation, treated with percutaneous transluminal coronary angioplasty-drug eluting system (PTCA-DES) on the right coronary artery, recurrence of ACS-STEMI August 2019 treated with PTCA-DES of anterior interventricular coronary artery. On March 28, onset of fever (39.8 °C), cough, dyspnea, asthenia; nasopharyngeal swab positive for the presence of SARS-CoV-2 virus RNA. On April 8, worsening of dyspnea and SO2 76%; hospitalization in the COVID-19 area. Non-invasive ventilation in CPAP for 10 days and subsequent high-flow O_2_-therapy. Discharged April 28, after nasopharyngeal swab was negative for the presence of SARS-CoV-2 virus RNA. On May 26, due to the persistence of asthenia, easy fatigue, dyspnea, cough, polymyalgia and mental confusion, the patient was hospitalized (Functional Status Scale 3). Laboratory parameters showed persistent inflammatory and procoagulative state (Erythrocyte Sedimentation Rate 46 mm; High Sensitivity C Reactive Protein 62.86 mg/L; Interleukin-6 53.45 pg/mL; D-dimer 1199.7 ng/mL; NT-ProBNP 1438.9 pg/mL). Bilateral peripheral areas of consolidation with surrounding ground-glass opacities were observed at CT scan (Figure 4).

The transthoracic echocardiographic study showed left ventricular dilatation, anterior hypokinesia and proximal interventricular septum and a reduction in left ventricular ejection fraction (Left Ventricular (LV) end diastolic dimension, 6.0 cm; LV end diastolic volume, 181 mL; LV end systolic dimension, 5.0 cm; LV end systolic volume, 119 mL; LV ejection fraction, 34%; Left atrial anteroposterior dimension, 4.5 cm; E/A ratio 0.72; PAPs, 19 mmHg). Therapy with ceftriaxone and enoxaparin was initiated and antihypertensive, hypoglycemic and cardioactive drugs were continued. After six days of therapy, the patient’s clinical condition and laboratory parameters improved (Erythrocyte Sedimentation Rate 28 mm; High Sensitivity C Reactive Protein 2.27 mg/L; Interleukin-6 2.45 pg/mL; D-dimer 824.65 ng/mL; NT-ProBNP 436.8 pg/mL). The patient was discharged with the following therapy: Pantoprazole 40 mg/die, Bisoprolol 2.5 mg/die, Rivaroxaban 2.5 mg/bid, acetylsalicylic acid 100 mg/die, Sacubitril/Valsartan 24 mg–26 mg/bid, Repaglinide 0.5 mg/die, Atorvastatin 40 mg/die. The Long-COVID syndrome is characterized, in addition to the persistence of symptoms typical of the acute phase, also by alterations in inflammatory and coagulation parameters [50]. Our patient showed concomitant procoagulative and inflammatory states following SARS-CoV-2 infection. The finding of a reduced LVEF and elevated levels of NT-ProBNP, as well as the persistence of symptoms, led to the therapeutic option of therapy with Sacubitril/Valsartan to counteract the progression of heart failure [51]. In our opinion, Sacubitril/Valsartan is also able to counteract the inflammatory storm in COVID-19 patients [52]. Our patient had a high cardiovascular residual risk, being hypertensive, diabetic, dyslipidemic, and having ischemic heart disease; furthermore, the condition of Long-COVID imposes an additional risk of major cardiovascular events [44] (Figure 5). Therefore, rivaroxaban therapy at vascular doses is mandatory [53].

## 9. Conclusions and Perspectives

Cardiovascular diseases affected most of the patients with COVID-19 admitted to hospitals, thus resulting in higher risk for in-hospital mortality [54]. DIC may also complicate the clinical course of COVID-19. Routine monitoring of hemostasis tests may be useful for guiding therapeutic approaches and preventing disease progression. Critically ill COVID-19 patients may show hypoxia or hemodynamic instability, which could be related to thromboembolic disease. The optimal thromboprophylactic regimen for patients hospitalized for COVID-19 is unknown [55,56]. Furthermore, the drug–drug interactions between some antiviral treatments and direct oral anticoagulants may favor the use of LMWH or unfractionated heparin in this population. Preclinical studies showed that rivaroxaban was able to contain the inflammatory response during virosis [35]. The advice is not to abruptly withdraw DOACs in high-risk patients as this pharmacological action may result in worsening clinical status and promoting adverse outcomes. We do believe that DOACs should be continued in patients with COVID-19 until further data are available. In Long-COVID patients, persistent inflammation and a prothrombotic state secondary to persistent endothelial damage requires careful monitoring and appropriate intervention.

## Figures and Tables

**Figure 1 viruses-13-01904-f001:**
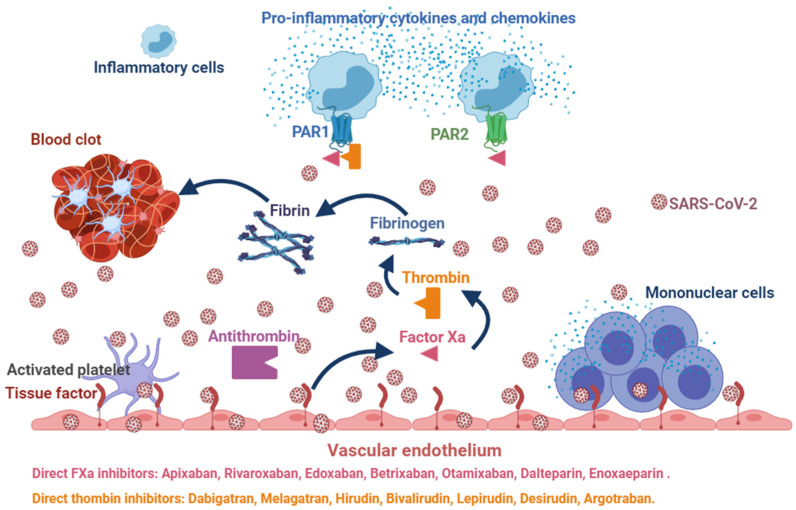
Interplay between SARS-CoV-2 infection, coagulation system, protease-activated receptors and inflammation. Activation of the coagulation cascade by the SARS-CoV-2 leads to the cleavage of fibrinogen into fibrin and platelet activation that can contribute to thrombosis. Platelet activation and fibrin degradation products can also enhance inflammation. Coagulation proteases can activate endothelial cells via protease-activated receptors that can increase the expression of inflammatory mediators. Drugs capable of inhibiting Factor Xa could attenuate the inflammatory response and modulate the coagulation cascade by reducing the formation of blood clots. This figure was created using the website https://app.biorender.com (accessed on 6 July 2021).

**Figure 2 viruses-13-01904-f002:**
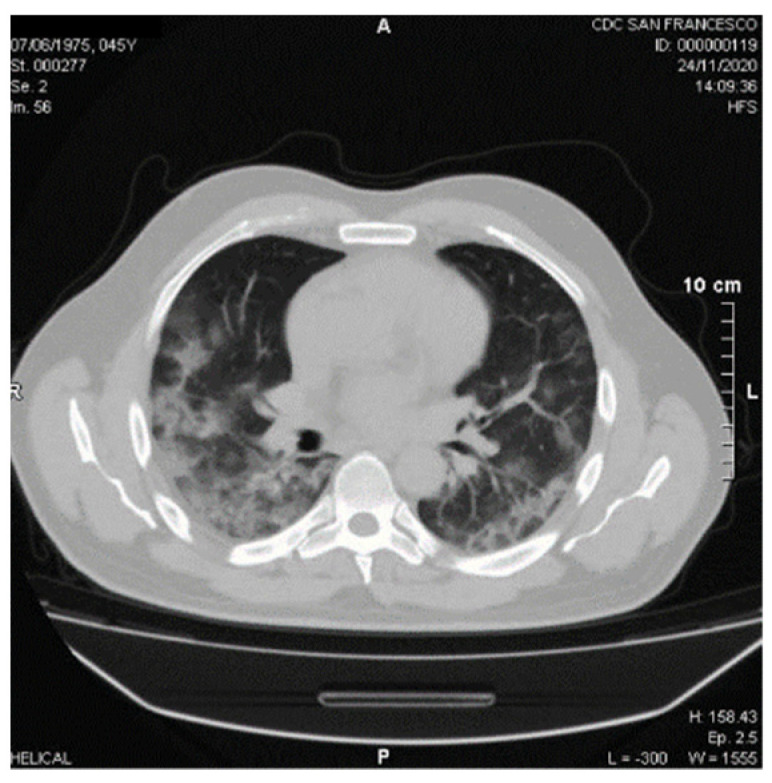
High resolution computed tomography (HRCT) chest, axial post-processed images, showing bilateral peripheral areas of ground-glass opacity (GGO).

**Figure 3 viruses-13-01904-f003:**
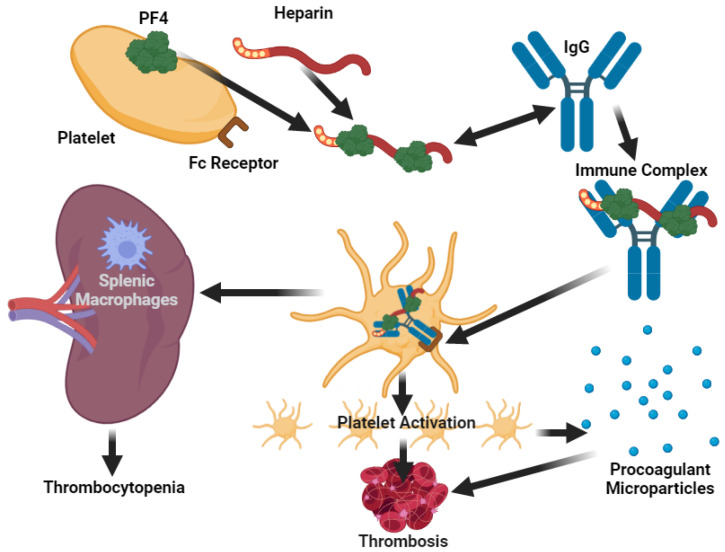
Pathophysiological mechanism of heparin-induced thrombocytopenia (HIT). Platelet factor 4 (PF4) is a chemokine secreted from the alpha-granules of platelets, released as tetramers. They bind to heparin and other proteoglycans and inactivate them. The binding of heparin to PF4 exposes new antigen sites and hence the formation of new (IgG) antibodies. Platelet Fc receptors bind the antibody-heparin-PF4, which contribute to thrombosis. Thrombocytopenia occurs by two mechanisms: removal of platelets with bound IgG by splenic macrophages and platelet consumption caused by thrombus formation. PF4 can also bind heparin sulfate on vascular endothelial cells; subsequent binding of the pathologic antibody to this PF4-heparin sulfate complex can injure the endothelium, which further promotes thrombosis. This figure was created using the website https://app.biorender.com (accessed on 12 July 2021).

**Figure 4 viruses-13-01904-f004:**
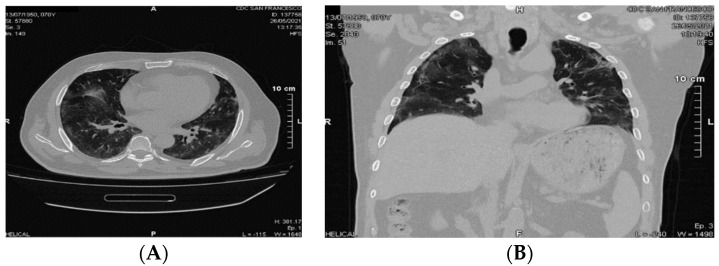
A 71-year-old male with sequel of COVID-19 pneumonia. Chest computed tomography (CT), axial image (**A**) showing bilateral peripheral areas of consolidation with surrounding ground-glass opacities (GGOs). Coronal image (**B**) showing bilateral peripheral area of interstitial thickening.

**Figure 5 viruses-13-01904-f005:**
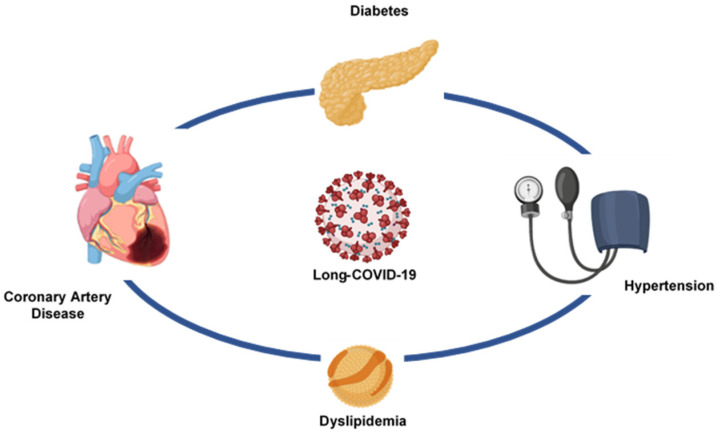
The cross-talk between coronary artery disease, diabetes, hypertension, dyslipidemia and Long-COVID-19.

**Table 1 viruses-13-01904-t001:** Summary of COVID-19 Patient Laboratory Data.

Laboratory Values (Reference Range)	24 November 2020	25 November 2020	26 November 2020	27 November 2020	28 November 2020	29 November 2020	30 November 2020
White Blood Cells count (3.7–10.3), ×10^9^/L	13.52	14.6	14.3	13.7	12.5	13.9	13.58
Neutrophils (40–75), %	87.6	88.0	87.2	86.8	81.0	82.1	77.2
Lymphocytes (19–48), %	6.6	6.0	6.5	9.2	10.2	10.6	11
Eosinophils (0–7), %	0	1	2	2	1	2	0.3
Red Blood Cells count (4.2–6.0), ×10^6^/L	5.32	5.42	5.12	4.92	4.91	5.2	5.31
Haemoglobin (13.7–17.5), g/dL	15.4	14.9	14.6	13.2	13.6	14.2	15.1
Platelet count (155–369), ×10^9^/L	311	70	90	180	220	310	346
Prothrombin time (9.6–12.5), second	13.4	18.2	16.2	13.2	10.2	10.2	10.4
International normalized ratio (INR) (0.9–1.2)	0.99	1.1	1.2	1.0	0.9	1.0	1.1
Activated partial thromboplastin time (19–30), s	29.3	33.2	34.6	35.1	29.1	28.5	27.6
Fibrinogen (150–450), mg/dL	570	220	300	420	510	480	366
Lactate dehydrogenase (140–280), U/L	1149	1520	1480	921	843	601	570
Creatinine (0.8–1.30), mg/dL	0.8	1.0	1.1	1.0	0.9	0.9	0.9
Erytrocite Sedimentation Rate (0–15), mm	62	121	144	80	73	52	31
High Sensitivity C Reactive Proteine (0–45), mg/L	104.9	158.8	161.2	82.1	40.1	18.2	2.23
IL-6 (0–6.4) pg/mL	36.74	84.2	96.8	72.3	42.1	16.3	5.56
D-dimer (250–500), ng/mL	1044	13,298	18,481	4280	3187	2128	347
Disseminated Intravascular Coagulation Score	0	6	6	4	1	0	0

**Table 2 viruses-13-01904-t002:** Clinical characteristics and baseline values of the study population.

Demographic, Medical History and Vital Signs	Long-COVID-19	No COVID-19
Number of patients, *n*	30	20
Sex, M/F, *n*	17/13	8/12
Age, years ^a^	58.6 ± 17.6	56.3 ± 14.7
Weight, kg ^a^	77.1 ± 14.5	73.8 ± 12
Height, cm ^a^	164.6 ± 11.4	169.1 ± 8.7
Body mass index, kg/m^2 a^	28.4 ± 4.2	25.7 ± 2.4
Pre-existing conditions in the last year, *n* (%)		
Cancer	2 (6.7%)	1 (5.0%)
Chronic heart disease	13 (43.3%)	6 (30.0%)
Chronic kidney disease	5 (16.6%)	2 (10.0%)
Chronic liver disease	3 (10.0%)	1 (5.0%)
Chronic lung disease	7 (23.3%)	7 (35.0%)
Chronic neurological disease	9 (30.0%)	5 (25.0%)
Diabetes	7 (23.7%)	3 (15.0%)
Hypertension	19 (63.3%)	11 (55.0%)
Mental health conditions	2 (6.66%)	1 (5.0%)
Obesity (Body Mass Index > 30)	11 (36.6%)	3 (15.0%)
Heart rate, bpm ^a^	73 ± 15	70 ± 13
Systolic arterial pressure, mmHg ^a^	121 ± 15	121 ± 17
Diastolic arterial pressure, mmHg ^a^	78 ± 12	76 ± 10
Therapies, *n* (%)		
ACE-I/ARB/ARNIs	19 (63%)	12 (60%)
Beta-blocker	11 (37%)	8 (40%)
ASA	13 (43%)	9 (45%)
Diuretics	11 (37%)	6 (30%)
Anticoagulants	12 (40%)	6 (30%)
Echocardiography Measurements		
LV end diastolic dimension, cm ^a^	4.8 ± 1	4.5 ± 0.6
LV end diastolic volume, mL ^a^	114.6 ± 52.5	94.1 ± 27.9
LV end systolic dimension, cm ^a^	3.2 ± 1.04	2.6 ± 0.5 *
LV end systolic volume, mL ^a^	48.7 ± 38.5	28 ± 10.5 ^†^
LV ejection fraction, % ^a^	61.9 ± 13.7	70.4 ± 5.7 •
Left atrial anteroposterior dimension, cm ^a^	3.7 ± 1.3	3.5 ± 0.5
E/A ratio ^a^	1.02 ± 0.4	1.1 ± 0.3
SPAP, mmHg ^a^	13.8 ± 10.5	14.6 ± 8.6

M = Male; F = Female; bpm = beats per minute; ACEi = angiotensin-converting enzyme inhibitor; ARB = angiotensin receptor blocker; ARNIs = Angiotensin Receptor Neprilysin Inhibitors; ASA= Acetylsalicylic Acid; LV = Left Ventricular; SPAP = Systolic Pulmonary Artery Pressure. ^a^ Mean ± standard deviation. * refers to *p* = 0.023; ^†^ refers to *p* = 0.024; • refers to *p* = 0.012.

**Table 3 viruses-13-01904-t003:** Laboratory data of the study population.

Laboratory Values (Reference Range)	Long-COVID-19	No COVID-19
White Blood Cells count (3.7–10.3), ×10^9^/L ^a^	6.84 ± 2.6	7.14 ± 2.3
Red Blood Cells count (4.0–10.0), ×10^6^/L ^a^	4.53 ± 0.6	4.8 ± 0.58
Haemoglobin (13.7–17.5), g/dL ^a^	14.9 ± 6.4	14.2 ± 1.8
Platelet count (155–369), ×10^9^/L ^a^	221 ± 92	244 ± 50
Prothrombin time (9.6–12.5), s ^a^	14.2 ± 2.5	13.5 ± 1.2
International normalized ratio (0.9–1.2) ^a^	1.07 ± 0.2	1.00 ± 0.09
Activated Partial Thromboplastin Time (19–30), s ^a^	30.6 ± 5.1	28.8 ± 2.6
Fibrinogen (150–450), mg/dL ^a^	364.8 ± 154.4	326.9 ± 86.1
Lactate dehydrogenase (140–280), U/L ^a^	448.1 ± 133	342.45 ± 90.5 *
Creatinine (0.8–1.30), mg/dL ^a^	0.92 ± 0.25	0.86 ± 0.23
Aspartate Aminotrasferase (0–31), U/L ^a^	25.04 ± 12.2	21.6 ± 12.2
Alanine Aminotrasferase (0–34), U/L ^a^	25.2 ± 14.5	20.9 ± 14.6
High Sensitivity C Reactive Protein (0–45), mg/L ^a^	16.3 ± 50.1	3.95 ± 8.8
Sodium (135–155), mEq/L ^a^	139 ± 2.7	139 ± 2.02
Potassium (3.5–5.5), mEq/L ^a^	4.1 ± 0.27	4.3 ± 0.4
D-dimer (250–500), ng/mL ^a^	1044.4 ± 1022	273.7 ± 106 ^†^
Erythrocyte Sedimentation Rate (0–15), mm ^a^	25.7 ± 33.2	15.5 ± 17.2
Albuminuria (0–2.5), mg/dL ^a^	120.7 ± 134.7	64.6 ± 17.7
Interleukin-6 (0–6.4), pg/mL ^a^	13.2 ± 3	3 ± 2.7 •
High-sensitivity Cardiac Troponin (<19), ng/mL ^a^	9 ± 26.3	1.6 ± 0.3
NT-ProBNP (<450), pg/mL ^a^	587.4 ± 273	273.5 ± 147.9 ^◊^
SARS-CoV-2 Anti-Spike IgM (<1), EU/mL ^a^	12.2 ± 35.5	1.04 ± 2.4
SARS-CoV-2 Anti-Spike IgG (<10), EU/mL ^a^	91.5 ± 130.1	35.9 ± 61.5
Serum Ferritin (20–300), ng/mL ^a^	144.6 ± 158.6	113 ± 85.7

^a^ Mean ± standard deviation; * refers to *p* = 0.004; ^†^ refers to *p* = 0.002; • refers to *p* = 0.024; ^◊^ refers to *p* < 0.0001.

## Data Availability

The dataset generated and analyzed during the current study is available from Domenico Acanfora, Department of Internal Medicine San Francesco Hospital, Viale Europa 21, 82037 Telese Terme, Benevento, Italy, E-mail: domenico.acanfora29@gmail.com.

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
