# Peer review of "The Cross-Talk between Thrombosis and Inflammatory Storm in Acute and Long-COVID-19: Therapeutic Targets and Clinical Cases"

_viruses, 2021, doi:10.3390/v13101904_

Round 1
Reviewer 1 Report
Summary:
The manuscript titled “The Cross-Talk Between Thrombosis and Inflammatory Strom In Acute and Long-COVID-19: Therapeutic Targets. Clinical Cases” reports the use of anticoagulants for acute and chronic COVID disease. The report demonstrates the use of these drugs in one clinical case. In general the manuscript is well-written and detailed. The background information on previous treatments and the argument to implement the use of anticoagulants like DOACS or LMWH is well described. All figures and tables are easy to read and supportive of the narrative in the manuscript. Although only one case was described in this report, future wide-spread use of particular anticoagulants may demonstrate the potential benefit of this treatment. I have the following minor comments:
TITLE
Is the period between “Therapeutic Targets. Clinical Cases” a typo? Please revise the sub title.
ABSTRACT
Line 21: Remove the extra space between “survivors, characterized…”.
Line 42: Does the term PF4 need to be defined?
INTRODUCTION
Lines 98-99: Please provide a reference for the statement that the death rate for SARS2 was double of SARS1 and MERS.
Lines 111: Is “bat” a typo? Please clarify.
CLINICAL CASE
Line 253: Remove the extra space between “[43]. A…”
Line 337: Add a hyphen to SARS-CoV2 to be consistent throughout the manuscript.
Author Response
Reviewer #1
We thank this Reviewer for the constructive comments and suggestions. Furthermore, we would like to really thank him/her for his/her appreciation about our research in the introduction section of his/her comments. This is our point-to-point reply.
TITLE
Is the period between “Therapeutic Targets. Clinical Cases” a typo? Please revise the subtitle.
Done.
We have changed the subtitle
ABSTRACT
Line 21: Remove the extra space between “survivors, characterized…”.
Done.
We have removed the extra space between “survivors, characterized…”.
Line 42: Does the term PF4 need to be defined?
Done.
We defined "for platelet factor 4 (PF4)"
INTRODUCTION
Lines 98-99: Please provide a reference for the statement that the death rate for SARS2 was double of SARS1 and MERS.
Done.
We have added the reference [16] and adapted the numbering of the references in the text and in the References section.
Casucci, G.; Acanfora, D.; Incalzi, R.A. The Cross-Talk between Age, Hypertension and Inflammation in COVID-19 Patients: Therapeutic Targets. Drugs Aging 2020, 37(11):779-785. doi: 10.1007/s40266-020-00808-4. Epub 2020 Oct 21. PMID: 33084001; PMCID: PMC7575413.
Lines 111: Is “bat” a typo? Please clarify. No, it’s not a typo.
Done.
No, it’s not a typo.
CLINICAL CASE
Line 253: Remove the extra space between “[43]. A…”
Done.
We have removed the extra space between ““[44]. A…”.
Line 337: Add a hyphen to SARS-CoV2 to be consistent throughout the manuscript.
Done.
We have added to hyphen to SARS-CoV-2

Reviewer 2 Report
The manuscript reviews the current approaches for treating disseminated intravascular coagulation in Covid-19 patients. Two clinical cases are presented for discussing anticoagulants treatments.
The manuscript is quite informative and interesting. It would be helpful to include information from clinical trials regarding the treatment with anticoagulants during and after SARS-CoV-2 infection.
Author Response
Reviewer #2
We thank this Reviewer for her/his appreciating our paper. We thank this Reviewer for the constructive comments and suggestions. Furthermore, we would like to really thank him/her for his/her appreciation about our research in his/her comments. This is our point-to-point reply. The manuscript is quite informative and interesting. It would be helpful to include information from clinical trials regarding the treatment with anticoagulants during and after SARS-CoV-2 infection. Done We added reference [57]: ATTACC Investigators; ACTIV-4a Investigators; REMAP-CAP Investigators, Lawler, P.R.; Goligher, E.C.; Berger, J.S.; et al. Therapeutic Anticoagulation with Heparin in Noncritically Ill Patients with Covid-19. N Engl J Med 2021,385(9):790-802. doi: 10.1056/NEJMoa2105911. Epub 2021 Aug 4. PMID: 34351721; PMCID: PMC8362594.
As far as we know, there are no trials with anticoagulants in post-COVID-19 patients. A trial was started in post-COVID-19 patients "Medically Ill hospitalized Patients for COVID-19 Thrombosis Extended ProphyLaxis with rivaroxaban ThErapy: Rationale and Design of the MICHELLE Trial" available online Journal Pre-Proof, not yet published.
